# Orthostatic tolerance and training methodology in physically active men and women

**Hayden W. Hess**[1]*, **Courtney E. Wheelock**[1,2], **David Hostler**[1,3]

1 Center for Research and Education in Special Environments (CRESE), Department of Exercise and Nutrition Sciences, University at Buffalo, Buffalo, New York, United States of America, 2 Department of Molecular Pharmacology and Physiology, University of South Florida, Tampa, Florida, United States of America, 3 Department of Emergency Medicine, University at Buffalo, Buffalo, New York, United States of America

* haydenhe@buffalo.edu

## Abstract

Previous studies suggest that training methodology alters orthostatic tolerance in elite and/or well-trained athletes. However, little is known about the effect of training methodology on orthostatic tolerance among the general physically active population. We tested the hypothesis that men and women participating in hybrid training (i.e., combined resistance and endurance training) would demonstrate superior orthostatic tolerance compared to aerobic endurance trained and recreationally active individuals. Twenty-nine participants were classified into one of the three groups depending on their current, self-reported training methodology. All participants reported at least 150 minutes per week of recreational activity (Recreationally Active Group; n = 10), hybrid training classes (Hybrid Group; n = 9), or endurance training (Endurance Group; n = 10) for ≥6 months. Anthropometrics (height and mass) were measured, and body composition was assessed via air displacement plethysmography. Orthostatic tolerance was assessed by a progressive lower-body negative pressure (LBNP) test and quantified via cumulative stress index (CSI). CSI did not differ between groups (p = 0.2542). Fat-free mass (FFM) was positively related to CSI ($R^2$ = 0.4092) and differed between groups (recreationally active: 56.1 ± 10.0 kg; hybrid: 65.5 ± 13.2 kg; endurance: 51.2 ± 10.5 kg; p = 0.0453). However, CSI normalized to FFM did not differ between groups (p = 0.6210; FFM missing for two participants). Among recreationally active adults, training methodology does not appear to modify orthostatic tolerance. Consistent with previous work, we report that body composition and anthropometrics are related to orthostatic tolerance highlighting the importance in maintaining lean mass regardless of training methodology.

## Introduction

Orthostatic stress is utilized to examine orthostatic tolerance and the control of blood flow, heart rate, and blood pressure regulation [1]. Orthostatic intolerance, and

**Data availability statement:** All relevant data are within the manuscript.

**Funding:** The author(s) received no specific funding for this work.

**Competing interests:** The authors have declared that no competing interests exist.

associated symptoms, can increase the risk of syncope and/or be used to diagnose orthostatic-related disorders (e.g., vasovagal syncope, orthostatic hypotension, and postural orthostatic tachycardia syndrome). In various populations with orthostatic intolerance, exercise and physical activity may be a viable intervention to improve or maintain orthostatic tolerance [2].

Indeed, previous work has demonstrated greater orthostatic tolerance in elite and well-trained power- and weightlifting athletes [3]. In contrast, endurance athletes have shown lower orthostatic tolerance likely due to increased vascular compliance and lower total and fat free mass compared to resistance trained individuals [4,5]. In both instances, the previous studies were conducted in highly trained individuals. It is not known if those results are applicable to a recreationally trained population. Another important consideration is the role of anthropometrics and body composition in orthostatic intolerance. This likely drives the sex differences observed in tolerance to orthostatic stress between men and women [6] and may explain the greater propensity for orthostatic-related disorders in women [7]. It is important to determine the role of training methodology in the general physically active population. Doing so could provide empirical evidence to generate guidance on training methodology selection for individuals with orthostatic intolerance.

Thus, this pilot study served two purposes with the overall aim to identify potential targets to improve orthostatic tolerance among the general population for future prospective exercise intervention studies. First, we aimed to examine how training methodology affects orthostatic tolerance in physically active men and women. We hypothesized that men and women participating in hybrid training (i.e., combined resistance and endurance training) would demonstrate superior orthostatic tolerance to aerobic endurance and recreationally active individuals. Due to potential differences in anthropometric measures and body composition between individuals currently participating in distinct training methodologies, a secondary purpose was to explore the relation between arthrometric measures and body composition on orthostatic tolerance. We hypothesized that there would be a positive relation between CSI and anthropometric measures (e.g., height and weight) and/or lean body mass.

## Methods

This pilot study was approved by the Institutional Review Board at the University at Buffalo (IRB# 00002304), conformed to the Declaration of Helsinki, except registration in a database. The study was active between 8/3/2018 and 5/16/2024. Before participation, each participant was informed of the procedures and risks before providing informed written consent.

### Participants

A convenience sample of twenty-nine participants were recruited from the collegiate and surrounding community via study flyer, email list-serv, and word-of-mouth. Male and female participants were included in the study if they were aged 18–39 years, self-reported to be in good health, and met criteria of one training methodology (outlined below). Indeed, all participants were free of chronic disease (i.e.,

cardiovascular, cerebrovascular, metabolic, and/or neurological diseases) and regularly engaged in physical activity as reported via the International Physical Activity Questionnaire [IPAQ] [8]. Women were not pregnant, which was confirmed via urine pregnancy test prior to any experimental procedures, self-reported to be normally menstruating, and had no diagnosis of a menstrual cycle disorder. Because menstrual cycle phase does not affect orthostatic tolerance in healthy women [9], women were tested at any point during their menstrual cycle and neither cycle phase nor contraceptive use were recorded. Due to this being a pilot study, no a priori power calculation was completed to determine total and/or group sample sizes.

## Training methodology group classification

The participants were classified into one of three groups based on current training methodology. Minimum eligibility criteria across all groups were self-reported engagement in ≥150 minutes per week of physical activity for ≥6 months, confirmed using the International Physical Activity Questionnaire (IPAQ). Following this eligibility check, participants completed a health screening questionnaire that assessed training methodology and structure for group classification. The recreationally active group (n = 10) was designed to represent a general physically active population and included individuals engaging in non-structured exercise and/or recreational physical activities (e.g., outdoor recreation such as hiking, casual sport such as tennis or golf, recreational league sports, or mixed activities) rather than a single standardized training program. The hybrid group (n = 9) reported participation exclusively in structured hybrid training classes at a local gym. The endurance group (n = 10) reported participation exclusively in structured endurance activities (e.g., running, cycling, swimming, or triathlon training). Training history and group assignment were based on self-report and were not objectively verified (e.g., training logs or wearable-derived volume/intensity). Study personnel reviewed responses with participants to confirm the reported training methodology and group classification prior to anthropometric, body composition, and orthostatic tolerance testing.

## Experimental protocol

Following consent and health screening, subjects completed anthropometric and body composition testing. Height was measured with a custom stadiometer (Pelstar LLC, McCook, IL) and body mass was measured with a weight scale (Scale-Tronics 5201, Welch-Allyn, Chicago, IL). Body composition was measured via air displacement plethysmography via BodPod (COSMED USA Inc., Chicago, IL). Orthostatic tolerance was then assessed by LBNP test [10]. Subjects assumed a supine position, with their lower body placed inside the LBNP chamber and sealed at the waist with a custom fitted neoprene skirt. Following 20 minutes supine rest, the LBNP chamber was decompressed to – 20 mmHg. Every three minutes, pressure was reduced by 20 mmHg until subject showed physiological or subjective signs of pre-syncope. Pre-syncope was identified by the onset of syncopal signs and symptoms, which included feeling faint, sustained nausea, rapid and progressive decreases in blood pressure resulting in sustained systolic blood pressure being < 80 mmHg and/or relative bradycardia accompanied by a narrowing of pulse pressure. Orthostatic tolerance was quantified via cumulative stress index (CSI), which is the sum of the product of LBNP level and the duration of each level until termination (e.g., 20 mmHg*3 min + 40 mmHg*3 min, etc.).

## Data and statistical analyses

Subject characteristics were compared across groups using one-way ANOVA with Tukey-adjusted post hoc comparisons when appropriate. Data are reported as mean ± SD. Summary data are also presented stratified by biological sex for transparency. However, the study was not powered a priori to examine sex-specific effects, and no formal analyses were performed. Between-group differences in cumulative stress index (CSI) during lower-body negative pressure (LBNP) were evaluated using nonparametric methods due to non-normality. Specifically, distributional assumptions were assessed by visual inspection of Q–Q plots and by testing residual normality (Shapiro–Wilk). $Log_{10}$ transformation did

not adequately normalize CSI residuals (Shapiro–Wilk: p = 0.0416). Therefore, CSI was compared across groups using a Kruskal–Wallis test. The relation between CSI and anthropometric/body composition measures were assessed using simple linear regression with best-fit line, 95% confidence bands, and $R^2$ reported. Because fat-free mass (FFM) differed between groups and was positively related to CSI in simple linear regression analysis, we conducted additional regression analyses to evaluate whether group differences persisted after accounting for FFM and to assess whether the relation between CSI and FFM differed by training group. Specifically, multiple linear regression models were fit including group and FFM, and group×FFM interaction terms were tested. For descriptive comparison of CSI relative to body size, CSI was also expressed per kg FFM. Residual normality for CSI/FFM was assessed and after $log_{10}$ transformation, residuals met normality assumptions (Shapiro–Wilk: p = 0.3193) and groups were compared using one-way ANOVA. All analyses were performed in GraphPad Prism (version 10.4.2). Statistical significance was set a priori at p ≤ 0.05. Where applicable, exact p-values and $R^2$ values are reported.

## Results

### Participant characteristics

Participant characteristics are summarized in Table 1. Age and height did not differ between groups (age: p = 0.1230; height: p = 0.2159). In contrast, body mass and BMI differed across groups (body mass: p = 0.0141; BMI: p = 0.0123),

**Table 1. Participant characteristics.**

| Parameter | Rec Active | Hybrid | Endurance | One-way ANOVA |
|---|---|---|---|---|
| Age, y | | | | |
| All: | 23 ± 1 | 27 ± 4 | 23 ± 0 | p = 0.1230 |
| Men: | 22 ± 2 | 24 ± 4 | 23 ± 5 | |
| Women: | 24 ± 1 | 30 ± 3 | 23 ± 5 | |
| Height, cm | | | | |
| All: | 168 ± 7 | 172 ± 6 | 166 ± 9 | p = 0.2159 |
| Men: | 173 ± 5 | 176 ± 9 | 173 ± 4 | |
| Women: | 163 ± 3 | 168 ± 9 | 159 ± 4 | |
| Body mass, kg | | | | |
| All: | 75 ± 2 | 77 ± 9@ | 64 ± 7 | *p = 0.0141* |
| Men: | 76 ± 7 | 83 ± 10 | 69 ± 5 | |
| Women: | 60 ± 5 | 71 ± 12 | 59 ± 6 | |
| BMI, kg/m² | | | | |
| All: | 24 ± 2 | 26 ± 1@ | 23 ± 0 | *p = 0.0123* |
| Men: | 25 ± 2 | 27 ± 2 | 23 ± 2 | |
| Women: | 22 ± 2 | 25 ± 2 | 23 ± 2 | |
| Body Fat, % | | | | |
| All: | 17 ± 4 | 17 ± 8 | 18 ± 11 | p = 0.8104 |
| Men: | 14 ± 5 | 11 ± 5 | 11 ± 5 | |
| Women: | 20 ± 3 | 23 ± 5 | 26 ± 3 | |
| Fat free mass, kg | | | | |
| All: | 56 ± 12 | 65 ± 16@ | 52 ± 13 | *p = 0.0453* |
| Men: | 65 ± 5 | 76 ± 5 | 61 ± 6 | |
| Women: | 48 ± 3 | 54 ± 7 | 43 ± 5 | |

Data are reported as mean±SD. @ different from endurance group (p < 0.05).

whereas percent body fat did not differ (p = 0.8104). Fat-free mass (FFM) also differed between groups (p = 0.0453) and was higher in the hybrid group (65 ± 16 kg) than the endurance group (52 ± 13 kg).

## Orthostatic tolerance by training group

Orthostatic tolerance, quantified as cumulative stress index (CSI) during lower-body negative pressure (LBNP), did not differ across recreationally active, hybrid, and endurance-trained participants (p = 0.2542; Fig 1A). CSI normalized to FFM (CSI/FFM) similarly did not differ between groups (p = 0.6210; Fig 1B).

**A**

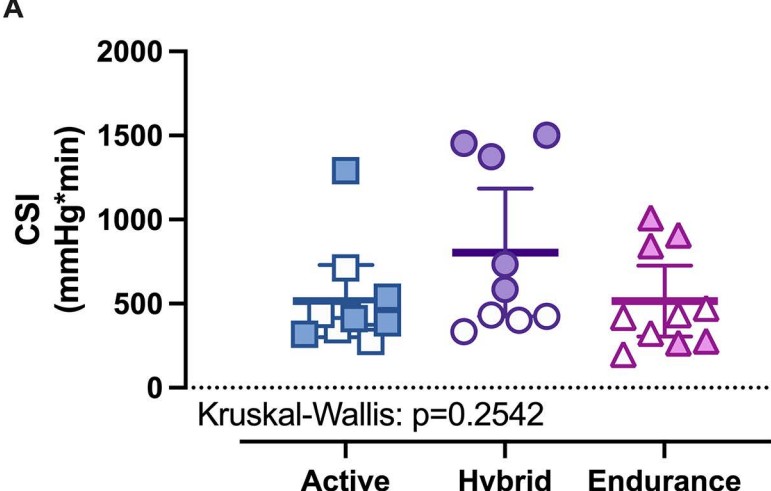

**B**

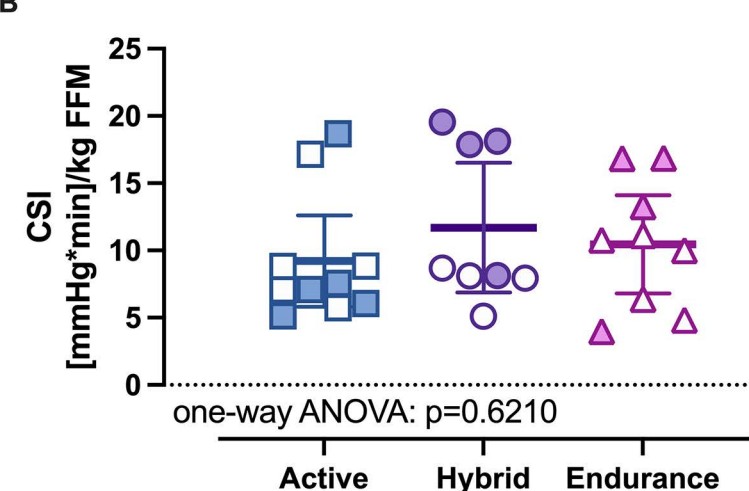

**Fig 1. Cumulative stress index (CSI) during LBNP by training group.** (A) Absolute CSI in recreationally active (n = 10), hybrid (n = 9), and endurance-trained participants (n = 10). Between-group differences were assessed using a Kruskal–Wallis test. (B) CSI normalized to fat-free mass (CSI/FFM; mmHg/min/kg) in recreationally active (n = 10), hybrid (n = 8), and endurance-trained participants (n = 9). *Note: FFM was missing for one hybrid participant and one endurance participant. Between-group differences were assessed using one-way ANOVA on log10-transformed CSI/FFM. Individual data are shown (closed symbols: men; open symbols: women; squares: recreationally active; circles: hybrid; triangles: endurance). Exact p-values are shown in each panel.

### Relation between CSI and anthropometric/body composition measures

In simple linear regression, CSI was positively related with FFM ($R^2 = 0.4092$; $p = 0.0003$; Fig 2D). Relations between CSI and additional anthropometric/body composition measures are shown in Fig 2.

### Multiple linear regression

To address between-group differences in FFM, we fit multiple linear regression models including training group, FFM, and group×FFM interaction terms. In this model, neither the main effects nor the interaction terms were significant (overall model: $p = 0.3561$; Hybrid×FFM: $p = 0.8239$; Endurance×FFM; $p = 0.3416$), indicating no evidence that the CSI–FFM relation differed by training group.

## Discussion

The purpose of the present study was to examine whether training methodology is associated with orthostatic tolerance in physically active adults. We hypothesized that individuals engaged in hybrid training would demonstrate superior orthostatic tolerance compared with endurance-trained and recreationally active individuals. Contrary to this hypothesis, the primary finding was that orthostatic tolerance did not differ across training groups. This conclusion was consistent across statistical approaches, including comparison of absolute CSI, CSI normalized to fat-free mass (CSI/FFM), and multiple linear regression analyses that accounted for between-group differences in FFM.

In exploratory analyses, CSI demonstrated a positive relation with anthropometric and body composition measures, with fat-free mass showing the strongest relation with CSI. However, when models incorporated both training group and FFM, neither variable independently predicted CSI, suggesting that within this physically active cohort, body composition and training classification share variance and that no single variable appeared as an independent determinant of CSI.

Prior work in highly trained endurance athletes has reported reduced orthostatic tolerance relative to untrained or resistance-trained individuals, with altered vascular compliance and related hemodynamic adaptations proposed as contributing mechanisms [11]. In the present study, participants represent a general physically active population rather than elite athletes, and the training volumes and physiological adaptations associated with high-level endurance training may not be fully represented. Accordingly, the absence of detectable group differences may reflect that broad training methodology labels (i.e., recreationally active, hybrid, endurance-trained) are insufficient to separate the specific physiological adaptations that meaningfully influence orthostatic tolerance in non-elite participants.

We also present data stratified by biological sex. Although the study was not designed or powered to test sex-specific effects, stratification does not appear to alter the overall inference of no training-group differences in CSI. That said, previous work has demonstrated sex differences in orthostatic tolerance [6] and that women have a higher prevalence of orthostatic-related conditions (e.g., orthostatic hypotension, postural orthostatic tachycardia syndrome [POTS]) [7]. However, the present data do not permit strong conclusions regarding whether physical activity attenuates sex-related differences in orthostatic tolerance. Larger studies designed a priori to evaluate sex as an effect modifier will be needed to address this more rigorously.

### Limitations

This study has several limitations that inform interpretation. First, the absence of a sedentary control group limits inference regarding the effect of exercise training per se on orthostatic tolerance, independent of training modality. Accordingly, the present comparisons are restricted to differences among physically active groups and cannot determine whether CSI during LBNP differs between trained and untrained individuals. However, previous work has demonstrated that exercise training does improve orthostatic tolerance across a range of populations [12,13]. Second, training modality and history were determined by self-report. Although minimum physical activity criteria were confirmed using the IPAQ and study

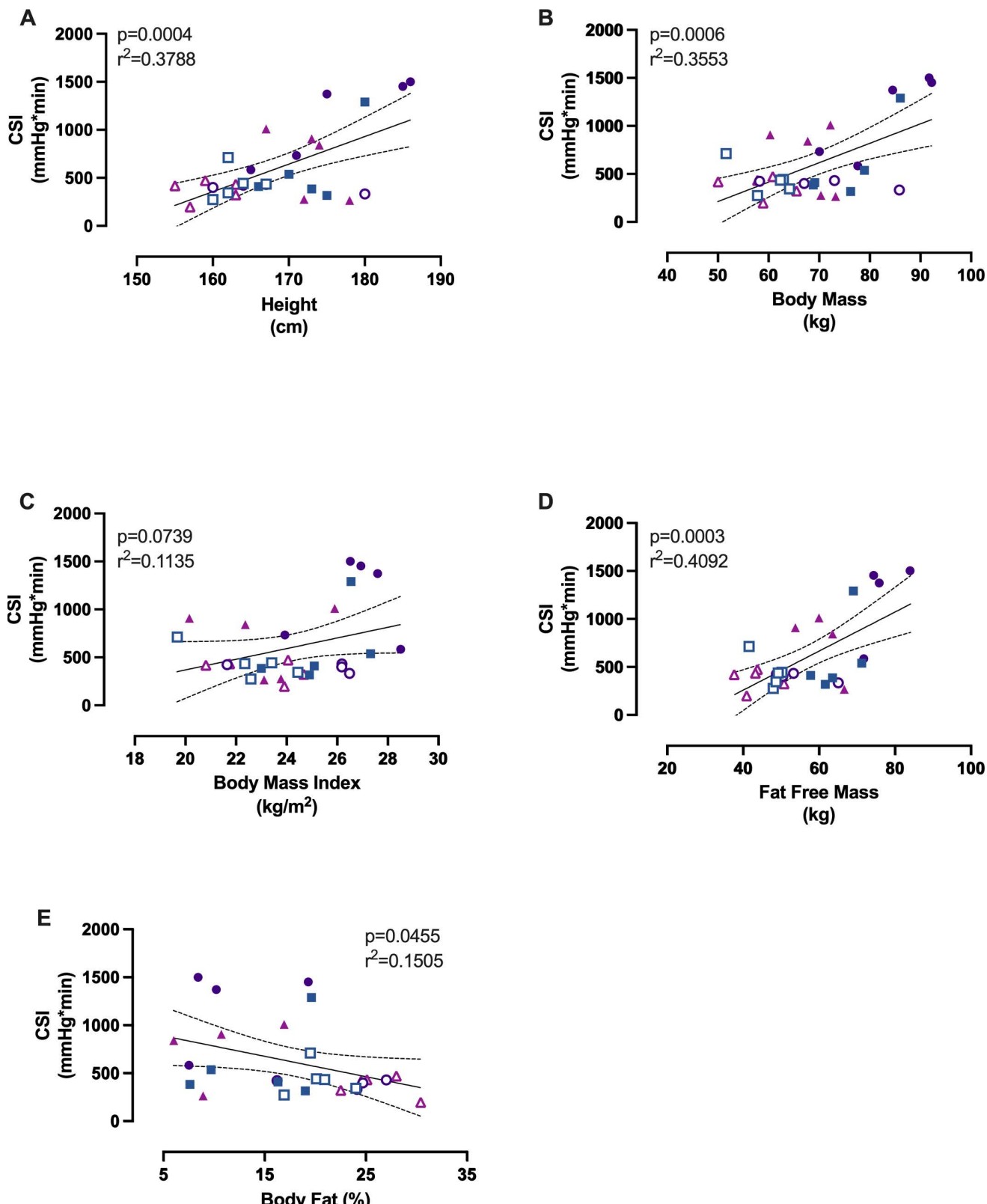

**Fig. 2. Relation between cumulative stress index (CSI) during LBNP and anthropometric/body composition measures.** Figures depict the relations between CSI and (A) height, (B) body mass, (C) body mass index, (D) fat-free mass, and (E) percent body fat. Relations were assessed using

simple linear regression. Solid lines indicate the best-fit regression line and dashed lines indicate the 95% confidence band. Individual data are shown (closed symbols: men; open symbols: women; squares: recreationally active; circles: hybrid; triangles: endurance). Exact p-values and R² are shown in each panel.

personnel reviewed responses with participants prior to testing, training volume, intensity, and adherence were not objectively verified (e.g., training logs or wearable-derived metrics). Third, fitness-related characteristics (e.g., aerobic capacity or maximal strength) were not directly measured. Because these variables may contribute to orthostatic tolerance, unmeasured differences in fitness could explain additional variance in CSI beyond the anthropometric and body composition measures reported. Fourth, groups were not randomized and differed in body size and composition (including FFM), raising the possibility of confounding in between-group comparisons. To address this concern, we used a nonparametric primary group comparison for CSI and conducted multiple linear regression analyses that accounted for FFM and tested group-by-FFM interaction terms. That said, the study was not designed to establish causal effects of training modality on orthostatic tolerance. Finally, a resistance-training-only group was not included. This limits inference regarding the independent contribution of resistance training and constrains interpretation of training-specific effects relative to a purely resistance-trained group. Future studies incorporating a resistance-only group and objective verification of training exposure would strengthen internal validity and mechanistic interpretation.

### Practical applications

Exercise is a viable method for enhancing orthostatic tolerance [14]. Indeed, in a position statement [15] it is recommended that individuals with orthostatic-related conditions or that are symptomatic during orthostatic stress (e.g., sit-to-stand) engage in regular physical activity. The data showing the relationship between lean body mass and orthostatic tolerance indicates that maintenance, or development, of skeletal muscle is important so a combination of aerobic and resistance training that result in cardiovascular adaptations that improve venous return and strength may be optimal. These adaptions may help mitigate orthostatic hypotension and associated symptoms such as dizziness or syncope.

### Conclusion

In conclusion, among physically active adults, orthostatic tolerance assessed by CSI during LBNP did not differ between recreationally active, hybrid-trained, and endurance-trained individuals. While anthropometric and body composition measures, particularly FFM, were associated with CSI in unadjusted analyses, sensitivity analyses did not support training methodology as an independent determinant of orthostatic tolerance in this cohort. These findings suggest that within physically active populations, broad training-category labels may be less informative than individual physiological characteristics for explaining variability in orthostatic tolerance.

### Acknowledgments

We thank the participants for participating in our study. Additionally, we thank the research staff and other students for assistance during data collection.

### Author contributions

**Conceptualization:** Hayden W. Hess, Courtney E. Wheelock, David Hostler.

**Data curation:** Hayden W. Hess, Courtney E. Wheelock.

**Formal analysis:** Hayden W. Hess, Courtney E. Wheelock.

**Supervision:** David Hostler.

**Writing – original draft:** Hayden W. Hess, David Hostler.

**Writing – review & editing:** Hayden W. Hess, Courtney E. Wheelock, David Hostler.

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
