## [Decision Letter · Decision Letter 0]

21 Oct 2025

Orthostatic tolerance is not modified by training methodology in physically active men and women

PLOS ONE

Thank you for submitting your manuscript to PLOS ONE. After careful consideration, we feel that it has merit but does not fully meet PLOS ONE’s publication criteria as it currently stands. Therefore, we invite you to submit a revised version of the manuscript that addresses the points raised during the review process.

We look forward to receiving your revised manuscript.

Kind regards,

Manzur Kader, Ph.D

Academic Editor

PLOS ONE

Journal Requirements:

[No authors have competing interests].

Reviewers' comments:

Reviewer's Responses to Questions

**Comments to the Author**

1. Is the manuscript technically sound, and do the data support the conclusions?

Reviewer #1: Yes

Reviewer #2: Partly

Reviewer #3: Partly

Reviewer #4: Yes

2. Has the statistical analysis been performed appropriately and rigorously?

Reviewer #1: Yes

Reviewer #2: Yes

Reviewer #3: No

Reviewer #4: Yes

3. Have the authors made all data underlying the findings in their manuscript fully available?

Reviewer #1: Yes

Reviewer #2: Yes

Reviewer #3: Yes

Reviewer #4: Yes

4. Is the manuscript presented in an intelligible fashion and written in standard English?

Reviewer #1: Yes

Reviewer #2: Yes

Reviewer #3: Yes

Reviewer #4: Yes

Reviewer #1: The study is intriguing, and the manuscript is well written. Well done to the authors. However, I have a few minor comments that may help improve clarity and strengthen the manuscript.

Specific Comments:

Study Purpose

The stated purpose of the study focuses only on examining whether the training methodology affects orthostatic tolerance in physically active men and women. However, the manuscript also presents analyses on the relationship between the cumulative stress index and anthropometric/body composition measures. While these findings are informative and potentially significant, they are not reflected in the introduction. Please include this as part of the study objectives.

Participant Recruitment and Group Allocation

The manuscript does not provide details on how participants were recruited or how they were allocated into groups. It is unclear whether randomisation was applied or what sampling method was used. Please clarify.

Sample Size and Eligibility Criteria

The inclusion and exclusion criteria for participants are not described, nor is there an explanation for the relatively small sample size. This information would strengthen the transparency and reproducibility of the study.

Description of Training Groups

The description of the three groups (recreationally active, hybrid, and endurance) lacks sufficient detail. The specific training modalities included, along with their intensity, volume, and FITT principles, are unclear. Moreover, it is not specified whether training was home-based, self-directed, or prescribed/monitored by the researchers. A detailed explanation is crucial to allow readers to fully understand the intervention and its potential impact.

Reviewer #2: The study investigates the effect of different training methodologies (hybrid, endurance, and recreational activity) on orthostatic tolerance in physically active men and women. The study is relevant and timely, given the increasing recognition of orthostatic intolerance and its association with conditions like orthostatic hypotension, and postural orthostatic tachycardia syndrome (POTS). The study is particularly valuable because its scope traverse beyond elite athletes to a general physically active population. The introduction of the study effectively situates within existing literature, highlighting the gaps regarding non-elite and physically active individuals. Again, the hypothesis was clearly stated. Also, recruitment of participants from a college and surrounding community provides a mixed, realistic sample.

However:

1.The language use seemed too technical to comprehend by ordinary readers

2.The methods used not clearly defined making it difficult to understand the kind of methods which were applied in the current study and why. How was the sample size selected? Readers would want to know.

3. Only 29 participants were sampled across three groups, this limits statistical power, particularly for sex-stratified analysis.

4.The absence of a sedentary group makes it difficult to isolate the impact of any exercise vs. no exercise. This limits the interpretation of whether physical activity itself, rather than training modality, drives improvements.

5. Authors indicated that women were tested at any point in their menstrual cycle but ignored to mention whether they took into consideration or measures they took in controlling for hormonal fluctuations, which may have influenced orthostatic tolerance results.

Though the manuscript is well written, and with meaningful contributions to the field and to the understanding of orthostatic tolerance in the general physically active population, the study is methodologically weak. Authors are therefore strongly encouraged to work more to improve on the methods and the sample size for reconsideration.

Thank you

Reviewer #3: The investigation of orthostatic tolerance in physically active populations is of considerable clinical relevance, particularly given its implications for cardiovascular stability, autonomic function, and risk stratification in both athletic and general populations. Understanding how different training modalities influence orthostatic responses may inform exercise prescription and preventative strategies for syncope and related disorders.

While the manuscript addresses an important and underexplored area, it presents several formal and methodological limitations that warrant attention.

Firstly, the classification criteria for the "Recreationally Active" group are insufficiently defined. Unlike the hybrid and endurance groups, which are clearly delineated by training modality, the recreational group lacks operational clarity. It is unclear what types of activities qualify participants for inclusion. This ambiguity undermines the internal validity of group comparisons. Moreover, the absence of a resistance-only training group is a notable omission, particularly given the hypothesis centers on the comparative effects of hybrid training. Including such a group would have strengthened the design and allowed for more nuanced interpretation of training-specific effects.

Secondly, the presentation of participant characteristics within the Methods section is misplaced. Anthropometric and body composition data are outcome variables and should be reported in the Results section, where they can be contextualized and statistically interpreted. Additionally, the sample size (n=29) appears to have been selected arbitrarily, with no mention of a priori power analysis. This raises concerns regarding the statistical power to detect meaningful differences between groups, especially given the modest sample sizes per condition.

Another critical issue pertains to group comparability. The groups differ in fat-free mass, a variable shown to correlate positively with CSI. This imbalance introduces a confounding factor that may obscure true differences in orthostatic tolerance attributable to training methodology. Without appropriate statistical control or matching, conclusions drawn from between-group comparisons may be misleading. Furthermore, there appears to be an error in the reporting of p-values related to BMI. Clarification is needed regarding the statistical tests employed and whether assumptions of normality were verified. Although the manuscript mentions the use of means and standard deviations, it does not specify whether data distributions were assessed, nor whether parametric tests were appropriate.

The Methods section is presented as a continuous block of text, lacking clear subheadings such as "Participants," "Procedures," or "Data Analysis." This structure impairs readability and makes it difficult to locate specific methodological details.

Finally, the reliance on self-reported exercise data introduces potential bias. If training frequency and modality were not objectively verified, there is no guarantee that participants adhered to the reported routines with consistency. This limitation should be explicitly acknowledged, and its implications for data interpretation discussed.

In summary, while the study explores a clinically pertinent topic, its methodological shortcomings—particularly in group definition, sample size justification, data presentation, and statistical rigor—limit the strength of its conclusions. Addressing these issues would substantially enhance the manuscript’s scientific validity and interpretative clarity.

Reviewer #4: I congratulate the authors on their dedicated work. Your article is generally well-written and flows nicely. However, I believe that revising certain parts would improve its readability and appeal.

The participants' training intensity and history were defined solely by self-reporting, which may partially weaken the internal validity of the study from a methodological perspective; therefore, it is recommended that this limitation be clearly emphasised in the Limitations section of the article. Furthermore, the failure to consider the menstrual cycle phases of female participants in the evaluation process should be discussed as an important factor that could affect autonomic nervous system responses and orthostatic tolerance levels. In addition, the physiological mechanisms of the findings obtained in the study should be addressed in more detail, particularly in terms of parameters such as vascular adaptation, plasma volume, and muscle mass. These additions will strengthen the discussion section, clarify the biological basis of the results, and solidify the study's position in the literature.

**Do you want your identity to be public for this peer review?** For information about this choice, including consent withdrawal, please see our Privacy Policy

Reviewer #1: No

Reviewer #2: No

Reviewer #3: No

Reviewer #4: **Yes:** Görkem Açar

---

## [Author Response · Author response to Decision Letter 1]

20 Feb 2026

We sincerely thank the reviewers for the thoughtful and thorough review of our manuscript. We have addressed each of the comments below when/where possible and this has been detailed below each comment in blue text. We believe the manuscript is undoubtably improved.

An overall change was to the Title. The directional title was revised to be descriptive. “Orthostatic tolerance and training methodology in physically active men and women”.

Reviewer #1: The study is intriguing, and the manuscript is well written. Well done to the authors. However, I have a few minor comments that may help improve clarity and strengthen the manuscript.

Specific Comments:

Study Purpose

The stated purpose of the study focuses only on examining whether the training methodology affects orthostatic tolerance in physically active men and women. However, the manuscript also presents analyses on the relationship between the cumulative stress index and anthropometric/body composition measures. While these findings are informative and potentially significant, they are not reflected in the introduction. Please include this as part of the study objectives.

We thank you for the comment. A secondary objective has been added to the introduction with regards to examining the relation between CSI and anthropometrics.

Participant Recruitment and Group Allocation

The manuscript does not provide details on how participants were recruited or how they were allocated into groups. It is unclear whether randomisation was applied or what sampling method was used. Please clarify.

This point has been clarified in the Participants and Training Methodology Group Classification subsections within the Methods.

Sample Size and Eligibility Criteria

The inclusion and exclusion criteria for participants are not described, nor is there an explanation for the relatively small sample size. This information would strengthen the transparency and reproducibility of the study.

Where possible, these details have been included in the Participants subsection in the Methods. There was no a priori sample size calculation as this was a pilot study. This has been clarified throughout the manuscript.

Description of Training Groups

The description of the three groups (recreationally active, hybrid, and endurance) lacks sufficient detail. The specific training modalities included, along with their intensity, volume, and FITT principles, are unclear. Moreover, it is not specified whether training was home-based, self-directed, or prescribed/monitored by the researchers. A detailed explanation is crucial to allow readers to fully understand the intervention and its potential impact.

A Training Methodology Group Classification subsection in Methods has been added. It aims to address these points to the best of our abilities. All missing pertinent and/or objective training verification data has been added to the limitations and should be a focus in future trials.

Reviewer #2: The study investigates the effect of different training methodologies (hybrid, endurance, and recreational activity) on orthostatic tolerance in physically active men and women. The study is relevant and timely, given the increasing recognition of orthostatic intolerance and its association with conditions like orthostatic hypotension, and postural orthostatic tachycardia syndrome (POTS). The study is particularly valuable because its scope traverse beyond elite athletes to a general physically active population. The introduction of the study effectively situates within existing literature, highlighting the gaps regarding non-elite and physically active individuals. Again, the hypothesis was clearly stated. Also, recruitment of participants from a college and surrounding community provides a mixed, realistic sample.

However:

1.The language use seemed too technical to comprehend by ordinary readers

We had important consideration of technical jargon and revised where possible throughout the manuscript.

2.The methods used not clearly defined making it difficult to understand the kind of methods which were applied in the current study and why. How was the sample size selected? Readers would want to know.

Based on the other reviewers’ comments, the methods section has been revised to improve readability.

3. Only 29 participants were sampled across three groups, this limits statistical power, particularly for sex-stratified analysis.

Indeed, 29 participants is a relatively lower sample size. Due to this comment and those of other reviewers, we have elected to remove the sex stratified analyses and more simply identified the biological sex using different symbols when presenting individual data.

4.The absence of a sedentary group makes it difficult to isolate the impact of any exercise vs. no exercise. This limits the interpretation of whether physical activity itself, rather than training modality, drives improvements.

Indeed, this is a potential limitation to the present study and has been outlined in the Discussion section. However, this has already been demonstrated in previous work. Our aim was to provide pilot data to support training methodology selection for future exercise intervention studies.

5. Authors indicated that women were tested at any point in their menstrual cycle but ignored to mention whether they took into consideration or measures they took in controlling for hormonal fluctuations, which may have influenced orthostatic tolerance results.

Thank you for the comment. Previous work (PMID: 16683068) has shown that menstrual cycle phase does not affect orthostatic tolerance in healthy women, so we chose not to control for testing in a specific cycle phase or control for contraceptive use. Any sex-specific data have also been outlined to be exploratory and serve as pilot data for future power analyses and/or studies.

Though the manuscript is well written, and with meaningful contributions to the field and to the understanding of orthostatic tolerance in the general physically active population, the study is methodologically weak. Authors are therefore strongly encouraged to work more to improve on the methods and the sample size for reconsideration.

While there are limitations to the methods employed in the study, we believe our revision based on your and the other three reviewers’ comments greatly improve the manuscript.

Reviewer #3: The investigation of orthostatic tolerance in physically active populations is of considerable clinical relevance, particularly given its implications for cardiovascular stability, autonomic function, and risk stratification in both athletic and general populations. Understanding how different training modalities influence orthostatic responses may inform exercise prescription and preventative strategies for syncope and related disorders.

While the manuscript addresses an important and underexplored area, it presents several formal and methodological limitations that warrant attention.

Firstly, the classification criteria for the "Recreationally Active" group are insufficiently defined. Unlike the hybrid and endurance groups, which are clearly delineated by training modality, the recreational group lacks operational clarity. It is unclear what types of activities qualify participants for inclusion. This ambiguity undermines the internal validity of group comparisons. Moreover, the absence of a resistance-only training group is a notable omission, particularly given the hypothesis centers on the comparative effects of hybrid training. Including such a group would have strengthened the design and allowed for more nuanced interpretation of training-specific effects.

We thank you for your comment. We have included additional information in the Training Methodology Group Classification subsection in Methods that addresses each of these points to the best of our ability. Additional groups (e.g., resistance training only) are considered in the Limitations section.

Secondly, the presentation of participant characteristics within the Methods section is misplaced. Anthropometric and body composition data are outcome variables and should be reported in the Results section, where they can be contextualized and statistically interpreted. Additionally, the sample size (n=29) appears to have been selected arbitrarily, with no mention of a priori power analysis. This raises concerns regarding the statistical power to detect meaningful differences between groups, especially given the modest sample sizes per condition.

Thank you for the suggestion. We have moved the characteristics to the Results section. There was no a priori sample size determination. Rather this was to serve has a pilot study given its underexplored area. This has been better framed in the Introduction.

Another critical issue pertains to group comparability. The groups differ in fat-free mass, a variable shown to correlate positively with CSI. This imbalance introduces a confounding factor that may obscure true differences in orthostatic tolerance attributable to training methodology. Without appropriate statistical control or matching, conclusions drawn from between-group comparisons may be misleading. Furthermore, there appears to be an error in the reporting of p-values related to BMI. Clarification is needed regarding the statistical tests employed and whether assumptions of normality were verified. Although the manuscript mentions the use of means and standard deviations, it does not specify whether data distributions were assessed, nor whether parametric tests were appropriate.

We appreciate the reviewer’s concern regarding group comparability and the potential for fat-free mass (FFM) to confound between-group comparisons of orthostatic tolerance. We agree that FFM differed across training groups and note that FFM was positively related to CSI. To address potential confounding directly, we performed additional analyses that statistically controlled for FFM and tested whether the CSI–FFM relation differed by group. In multiple linear regression models including training group and FFM, training group was not independently associated with CSI, and there was no evidence of a group×FFM interaction. Thus, although FFM correlates with CSI in unadjusted analyses, our primary finding remains unchanged: we did not detect differences in orthostatic tolerance between training methodologies after accounting for body composition, and the CSI–FFM association does not appear to differ by training group in this cohort.

We also agree that the manuscript should more clearly describe the statistical tests employed and the evaluation of assumptions. We have revised the Methods to explicitly state that distributions and regression diagnostics were assessed (Q–Q plots and Shapiro–Wilk tests of residuals). Because CSI residuals remained non-normal after log10 transformation, the primary group comparison for CSI was conducted using a Kruskal–Wallis test. For CSI normalized to FFM, residual normality was met after log10 transformation and a one-way ANOVA was used. We have clarified in the Methods and figure legends which tests were used for each analysis and report exact p-values and R² where applicable.

The Methods section is presented as a continuous block of text, lacking clear subheadings such as "Participants," "Procedures," or "Data Analysis." This structure impairs readability and makes it difficult to locate specific methodological details.

Thank you for the comment. These subsections have been added to the Methods section.

Finally, the reliance on self-reported exercise data introduces potential bias. If training frequency and modality were not objectively verified, there is no guarantee that participants adhered to the reported routines with consistency. This limitation should be explicitly acknowledged, and its implications for data interpretation discussed.

This point is very well taken. We have included additional text in the Limitations subsection.

In summary, while the study explores a clinically pertinent topic, its methodological shortcomings—particularly in group definition, sample size justification, data presentation, and statistical rigor—limit the strength of its conclusions. Addressing these issues would substantially enhance the manuscript’s scientific validity and interpretative clarity.

Reviewer #4: I congratulate the authors on their dedicated work. Your article is generally well-written and flows nicely. However, I believe that revising certain parts would improve its readability and appeal.

The participants' training intensity and history were defined solely by self-reporting, which may partially weaken the internal validity of the study from a methodological perspective; therefore, it is recommended that this limitation be clearly emphasized in the Limitations section of the article.

In agreement with the other reviewers’ comments, we have included additional information regarding the training methodology groups and associated limitations with our recruitment methods in the Limitations subsection.

Furthermore, the failure to consider the menstrual cycle phases of female participants in the evaluation process should be discussed as an important factor that could affect autonomic nervous system responses and orthostatic tolerance levels.

Thank you for the comment. This is like that of Reviewer 2. Previous work (PMID: 16683068) has shown that menstrual cycle phase does not affect orthostatic tolerance in healthy women, so we chose not to control for testing in a specific cycle phase or control for contraceptive use.

In addition, the physiological mechanisms of the findings obtained in the study should be addressed in more detail, particularly in terms of parameters such as vascular adaptation, plasma volume, and muscle mass.

Indeed, others that have shown differences either 1) from training adaptation or 2) between two different groups (e.g., endurance vs. strength trained) have identified potential adaptive mechanisms (e.g., vascular compliance, hematological adaptations). However, we did not identify differences between groups. We have included some consideration of this point in the Discussion, particularly regarding that lack of sedentary control group (any exercise vs. no exercise).

These additions will strengthen the discussion section, clarify the biological basis of the results, and solidify the study's position in the literature.

---

## [Editor Report · Decision Letter 1]

3 Mar 2026

Orthostatic tolerance and training methodology in physically active men and women

PONE-D-25-45473R1

Dear Dr. Hayden,

We’re pleased to inform you that your manuscript has been judged scientifically suitable for publication and will be formally accepted for publication once it meets all outstanding technical requirements.

Kind regards,

Manzur Kader, Ph.D

Academic Editor

PLOS One